# APPROXIMATION OF THE GOMPERTZ TREND WITH A MULTILOGISTIC FUNCTION CONFERENCE SUBMISSIONS

## ABSTRACT

The paper deals with the comparison of the Gompertz function and the logistic function. We show that the Gompertz trend can be approximated with high accuracy by a sum of three logistic functions (multilogistic function). Two of them are increasing, and one is decreasing. We use second-order logistic wavelets to estimate the parameters of the multilogistic function.

Keywords: Gompertz function, logistic function, logistic mother wavelet, scalogram.

2020 Mathematics Subject Classification: 42C40, 65T60, 11B83

## 1 INTRODUCTION

The Gompertz function is described by the following autonomous differential equation of the first order

$$x'(t) = sx \log \frac{x_{sat}}{x} \qquad x(0) = x_0 > 0, \tag{1}$$

with parameters $s-$growth rate and $x_{sat}-$saturation level (asymptote), $0 < x_0 < x_{sat}$; $\log$ is the natural logarithm. After solving (1) we can write the Gompertz function in the following convenient form

$$x(t) = x_{sat} e^{-e^{-s(t-t_0)}}, \tag{2}$$

where constant $t_0$ appears in the integration process of (1) and is connected with the initial condition $x(0) = x_0 = x_{sat} e^{-e^{st_0}}$, thus $t_0 = \frac{1}{s} \log \log(x_{sat}/x_0)$. It is easy to check that $t_0$ is also the inflection point of $x(t)$ (2). Fig 1 shows an exemplary Gompertz funcion.

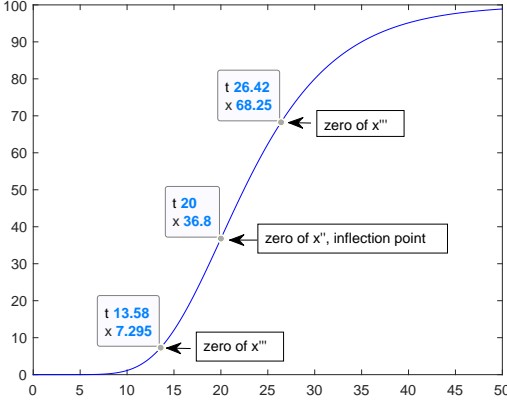

Figure 1: Exemplary Gompertz function with parameters $x_{sat} = 100, s = 0.15, t_0 = 20$

The Gompertz function (2) is the example of the so-called S-shaped curves and was first described and applied in actuarial mathematics in 1825 by Gompertz (1825). Since then, the Gompertz function

has found applications in probability theory (Gumbel distribution), biology, medicine, economics, engineering, physics and other fields. Many interesting applications of the Gompertz curve are given by Waliszewski & Konarski (2005).The first hundred years of the use of this function are well described by Winsor (1932). The interesting story of the next almost one hundred years can be found in the article by Tjørve & Tjørve (2017). In recent years, many papers have appeared in which the Gompertz function was used to describe the spread of COVID-19 pandemy (see Ohnishi et al. (2020), Dhahbi et al. (2022), Kundu et al. Kundu et al. (2021), Estrada & Bartesaghi (2022)).

The logistic equation defining the logistic function $x = x(t)$ has the form

$$x'(t) = \frac{s}{x_{sat}} \, x(x_{sat} - x), \quad x(0) = x_0. \tag{3}$$

where $t$ is time, and the parameters $s$-steepness or slope coefficient and $x_{sat}$-saturation level are real constants. We assume here that the saturation level can be a positive or negative number. The integral curve $x(t)$ of equation (3) satisfying the condition that $x(t)$ lies between zero and $x_{sat}$ is called the logistic function. The logistic function is used to describe and model various phenomena in physics, economics, medicine, biology, engineering, sociology and many other sciences. Logistic functions now seem even more important from the point of view of their possible applications, due to the theory of the Triple Helix (TH) developed in the 1990s by Etzkowitz & Leydesdorff (1995) (see also Leydesdorff (2021)). This theory explains the phenomenon of creating and introducing innovations under the influence of the interaction of three factors University-Industry-Government and relations between them. According to the TH theory, the phenomenon of the emergence of innovations can be described by means of logistic functions Ivanova (2022a), Ivanova (2022b) has shown that the KdV equation naturally appears in TH theory and has also applied it to other fields such as the COVID-19 pandemic or financial markets.

After solving the differential equation (3) we obtain the logistic function in the form

$$x(t) = \frac{x_{sat}}{1 + e^{-s(t-t_0)}}, \tag{4}$$

where $t_0$ is the inflection point associated with the initial condition $x(0) = x_0 = \frac{x_{sat}}{1 + e^{st_0}}$, then $t_0 = \frac{1}{s} \log\left(\frac{x_{sat} - x_0}{x_0}\right)$. At the point $t_0$, $x(t_0) = x_{sat}/2$. An exemplary logistic function is shown in the Fig 2.

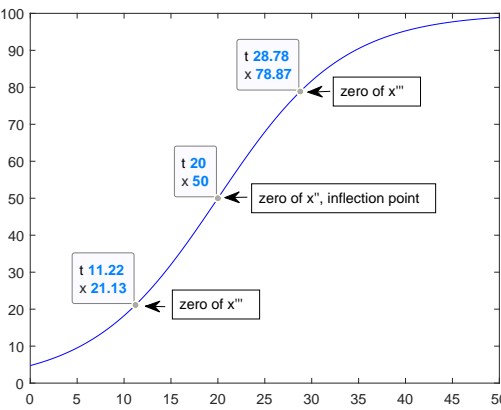

Figure 2: Exemplary logistic function with parameters $x_{sat} = 100, s = 0.15, t_0 = 20$

Mahjan et al. (1990) show areas in which various S-shaped curves are used as diffusion models. The authors find that the Bass function is useful for modeling of consumer durable goods, retail services, agricultural, education, and industrial innovations. The logistic curve serves as model in industrial, high technology, administrative innovations, and the Gompertz function can be used for modeling consumer durable goods and agriculture innovations.

Klein et al. (2017) use a basis function model (with several parametric forms) combined in a probabilistic prediction framework. They show that combining basis functions gives better predictions than simpler single-form models, especially when the observed part of the curve is not yet in the saturation regime.

The aim of this paper is to show that, for a given Gompertz function, it is possible to find some logistic functions (waves) such that their sum (multilogistic function) approximates the Gompertz function with high accuracy.

The structure of the article is as follows. In Sec. 2 we describe the basic properties of the second-order logistic wavelets. Sec. 3 is devoted to show connections between the Gompertz function and the logistic function. We prove that the Gompertz function can be approximated with high accuracy by a multilogistic function. The paper is concluded in Sec. 4.

## 2 LOGISTIC WAVELETS

We briefly outline the basic general properties of wavelets (cf. Daubechies (1992); Meyer & Ryan (1996); Meyer (1997)), which we will need later. A wavelet or mother wavelet (see Daubechies Daubechies (1992), p.24 ) is a function $\psi \in L^1(\mathbb{R})$ such that the following admissibility condition holds:

$$C_\psi = 2\pi \int_{-\infty}^{\infty} |\xi|^{-1} |\widehat{\psi}(\xi)|^2 d\xi < \infty, \tag{5}$$

where $\widehat{\psi}(\xi)$ is the Fourier transform $F(\psi)$ of $\psi$, i.e.,

$$F(\psi)(\xi) = \widehat{\psi}(\xi) = \frac{1}{\sqrt{2\pi}} \int_{-\infty}^{\infty} \psi(x) e^{-i\xi x} dx.$$

Since for $\psi \in L^1(\mathbb{R})$, $\widehat{\psi}(\xi)$ is continuous then condition (5) is only satisfied if $\widehat{\psi}(0) = 0$, which is equivalent to $\int_{-\infty}^{\infty} \psi(x) dx = 0$. On the other hand, Daubechies Daubechies (1992), p.24 points out that condition $\int_{-\infty}^{\infty} \psi(x) dx = 0$ together with a slightly stronger than the integrability condition $\int_{-\infty}^{\infty} |\psi(x)|(1 + |x|)^\alpha dx < \infty$, for some $\alpha > 0$ are sufficient for (5). Usually, in practice much more is assumed for the function $\psi$ hence, from a practical point of view, conditions $\int_{-\infty}^{\infty} \psi(x) dx = 0$ and (5) are equivalent. Suppose the function $\psi$ is also square-integrable, $\psi \in L^2(\mathbb{R})$ with the norm

$$||\psi|| = \left( \int_{-\infty}^{\infty} |\psi(x)|^2 dx \right)^{1/2}.$$

From a mother wavelet, one can generate a doubly-indexed family of wavelets (called children wavelets), by dilating and translating,

$$\psi^{a,b}(x) = \frac{1}{\sqrt{a}} \psi\left( \frac{x - b}{a} \right),$$

where $a, b \in \mathbb{R}$, $a > 0$. The normalization has been chosen so that $||\psi^{a,b}|| = ||\psi||$ for all $a, b$. In order to be able to compare different wavelets, it is convenient to normalize the wavelets, i.e., $||\psi|| = 1$.

The continuous wavelet transform (CWT) of a function $f \in L^2(\mathbb{R})$ for this wavelet family is defined as

$$(T^{wav} f)(a, b) = \langle f, \psi^{a,b} \rangle = \int_{-\infty}^{\infty} f(x) \psi^{a,b}(x) dx. \tag{6}$$

Rzadkowski & Figlia (2021) introduced the logistics wavelets of any order and then Rzadkowski (2023) presented their standardized form. The formula for the normalized second-order logistic mother wavelet $\psi_2(t)$ (see Fig. 3) is as follows

$$\psi_2(t) = \frac{\sqrt{30}}{1 + e^{-t}} \left( 1 - \frac{1}{1 + e^{-t}} \right) \left( 1 - \frac{2}{1 + e^{-t}} \right) = \frac{\sqrt{30}(e^{-2t} - e^{-t})}{(1 + e^{-t})^3}. \tag{7}$$

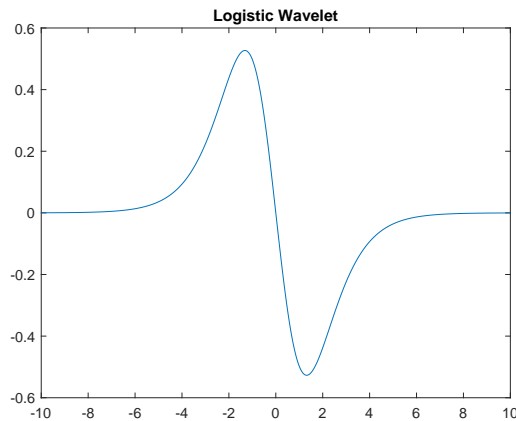

Figure 3: Logistic mother wavelet $\psi_2$

Formula (7) is simply the second derivative of the basic logistic function $x(t) = \dfrac{1}{1 + e^{-t}}$ multiplied by $\sqrt{30}$.

As usually, we generate from the mother wavelet a doubly-indexed family of wavelets from $\psi_2$ by dilating and translating

$$\psi_2^{a,b}(t) = \frac{1}{\sqrt{a}} \psi_2\Big(\frac{t-b}{a}\Big),$$

where $a, b \in \mathbb{R}, \ a > 0, \ n = 2, 3, \ldots$.

## 3 APPLICATIONS

Let $(y_n)$ be a time series. To calculate the CWT (Continuous Wavelet Transform) coefficients for the central second differences

$$\Delta^2 y_n = y_{n+1} - 2y_n + y_{n-1},$$

we use MATLAB's Wavelet Toolbox. Assume that the time series $(y_n)$ locally follows a logistic function $y(t) = \dfrac{y_{sat}}{1 + \exp(-\frac{t-b}{a})}$, i.e., $y_n \approx y(n) = \dfrac{y_{sat}}{1 + \exp(-\frac{n-b}{a})}$. By definition (7) we have

$$y''(t) = \frac{y_{sat}}{\sqrt{30}a^{3/2}} \psi_2^{a,b}(t).$$

**Lemma 1.** *The continuous wavelet transform CWT (6) of the function $y''(t)$, based on the logistic wavelets $\psi_2^{c,d}$*

$$(T^{wav} y'')(c,d) = \langle y'', \psi_2^{c,d} \rangle = \int_{-\infty}^{\infty} y''(t) \psi_2^{c,d}(t) dt,$$

*takes the maximum (for $y_{sat} > 0$) or minimum (for $y_{sat} < 0$) value when $c = a$ and $d = b$.*

*Proof.* Assume that $y_{sat} > 0$. By the Cauchy-Schwartz inequality

$$|(T^{wav} y'')(c,d)| = |\langle y'', \psi_2^{c,d} \rangle| \leq ||y''|| \ ||\psi_2^{c,d}|| = \frac{y_{sat}}{\sqrt{30}a^{3/2}} ||\psi_2^{a,b}|| \ ||\psi_2^{c,d}|| = \frac{y_{sat}}{\sqrt{30}a^{3/2}}.$$

However the maximum is reached for $c = a$, $d = b$, because:

$$(T^{wav} y'')(a,b) = \langle y'', \psi_2^{a,b} \rangle = \frac{y_{sat}}{\sqrt{30}a^{3/2}} \langle \psi_2^{a,b}, \psi_2^{a,b} \rangle = \frac{y_{sat}}{\sqrt{30}a^{3/2}}.$$

Similarly we consider the case $y_{sat} < 0$.

$\square$

In view of Lemma 1, for the maximal value of Index we get successively

$$\max(\text{Index}) = \sum_n \Delta^2 y_n \psi_2^{a,b}(n) \approx \sum_n \Delta^2 y(n) \psi_2^{a,b}(n) \approx \int_{-\infty}^{\infty} y''(t) \psi_2^{a,b}(t) dt$$

$$= \int_{-\infty}^{\infty} \frac{y_{sat}}{\sqrt{30} a^{3/2}} \psi_2^{a,b}(t) \psi_2^{a,b}(t) dt = \frac{y_{sat}}{\sqrt{30} a^{3/2}} \int_{-\infty}^{\infty} (\psi_2^{a,b}(t))^2 dt = \frac{y_{sat}}{\sqrt{30} a^{3/2}}. \quad (8)$$

Two parameters of a logistic wave, $b$ - shift (translation) and $a$ - dilation, can be read from the CWT scalogram by finding a point where the sum (8) (denoted in the scalogram by Index) is locally maximal (or locally minimal). It remains to determine the third parameter of the wave, i.e., its saturation level $y_{sat}$. Using (8) we can estimate the saturation level $y_{sat}$ as follows

$$y_{sat} \approx \sqrt{30} a^{3/2} \sum_n \Delta^2 y_n \psi_2^{a,b}(n) = \sqrt{30} a^{3/2} \max(\text{Index}). \quad (9)$$

Similarly we get

$$y_{sat} \approx \sqrt{30} a^{3/2} \sum_n \Delta^2 y_n \psi_2^{a,b}(n) = \sqrt{30} a^{3/2} \min(\text{Index}). \quad (10)$$

Consider the Gompertz function

$$x(t) = 100,000 \exp(-\exp(-\frac{t-50}{10})),$$

with parameters $x_{sat} = 100,000$; $t_0 = 50$; $s = 0.1$ and a time series $y_n$ following the same exact Gompertz growth trend (Fig. 4a)

$$y_n = x(n) = 100,000 \exp(-\exp(-\frac{n-50}{10})) \qquad n = 0, 1, 2, \ldots, 201. \quad (11)$$

Then we calculate the central first differences (Fig. 4b)

$$\Delta^1 y_n = (y_{n+1} - y_{n-1})/2, \quad n = 1, 2, \ldots 200,$$

and the central second differences (Fig. 4c)

$$\Delta^2 y_n = y_{n+1} - 2y_n + y_{n-1}, \quad n = 1, 2, \ldots, 200.$$

For the central second differences, we apply the MATLAB's cwt function, which uses second-order logistic wavelets (Fig. 4d). The marked point indicates the maximum of the Index. Using (9) we can estimate the parameters of the first logistic wave, which best fits the Gompertz trend $(y_n)$:

$$a = 6.115, \; b = 50, \; y_{sat} = \sqrt{30} \cdot 6.115^{3/2} \cdot 1062 = 87959.$$

Thus we obtain logistic function

$$h(t) = \frac{87959}{1 + e^{-\frac{t-50}{6.115}}}.$$

Then we repeat this procedure for the time series $(y_n - h(n))$, Fig. 5a. We calculate its first differences (Fig. 5b) and the second differences (Fig. 5c), for which we again apply the MATLAB's cwt function. (Fig. 5d) shows two points (the minimum and the maximum of Index) giving the other two logistic waves with the following parameters (for saturations levels we use (10) and (9)):

$$a = 4.028, \; b = 34, \; y_{sat} = \sqrt{30} \cdot 4.028^{3/2} \cdot (-246.4) = -10910,$$

and

$$a = 6.74, \; b = 66, \; y_{sat} = \sqrt{30} \cdot 6.74^{3/2} \cdot 226.4 = 21698.$$

The three logistic waves described above, when summed to a multilogistic function, approximate the time series $y_n$ with the maximum absolute error equal to 1808, with the root mean square error (RMSE) equal to 969 and $R^2 = 0.9998$. The RMSE error is mainly influenced by deviations, related to the mismatch of saturation levels, for values of $n$ greater than 100. The value of these deviations is approximately $100000 - 87959 + 10910 - 21698 = 1253$. After optimizing the parameters with

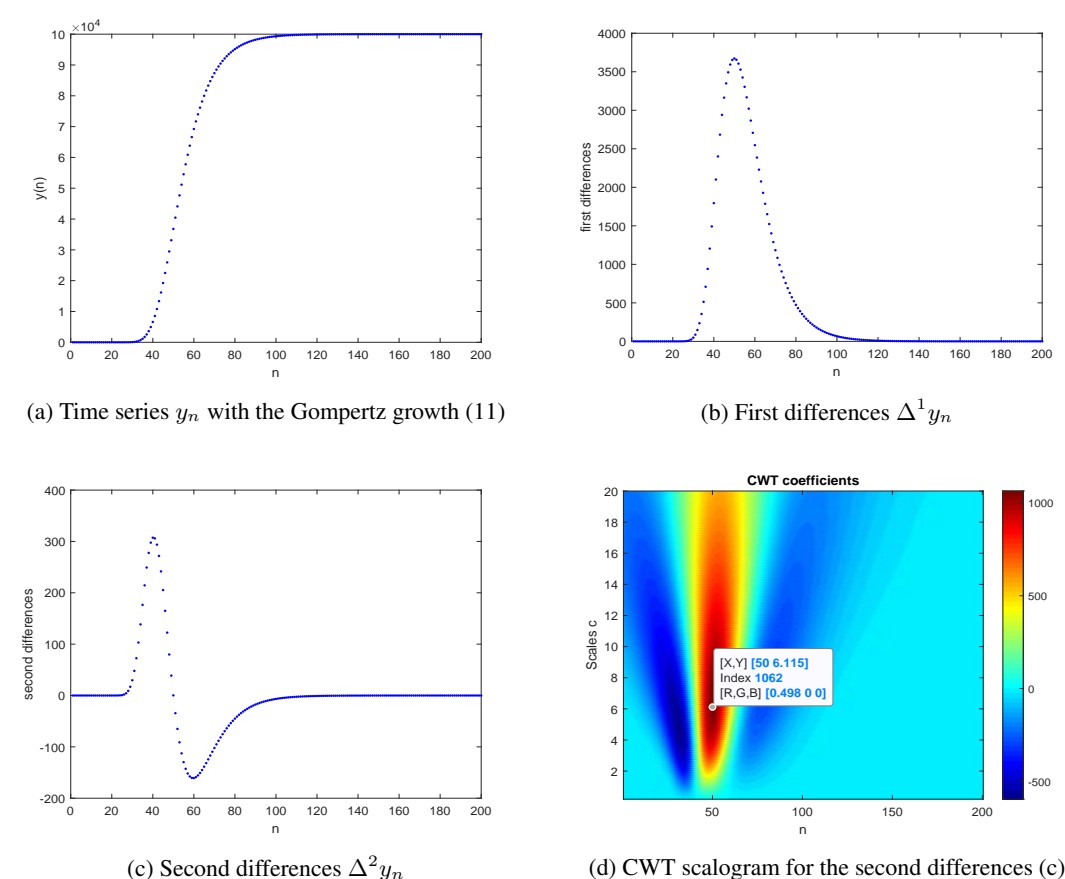

(a) Time series $y_n$ with the Gompertz growth (11)

(b) First differences $\Delta^1 y_n$

(c) Second differences $\Delta^2 y_n$

(d) CWT scalogram for the second differences (c)

Figure 4: Graphs and the CWT scalogram for the time series (11), with exact Gompertz growth

the objective function - minimization of the maximum absolute error - we get the multilogistic approximating function in the following form:

$$f(t) = \frac{88057}{1 + e^{-\frac{t-50}{6.17}}} - \frac{10919}{1 + e^{-\frac{t-33.55}{5.12}}} + \frac{22846}{1 + e^{-\frac{t-67.17}{8.77}}}. \tag{12}$$

The multilogistic function (12), $f(t)$ approximates the Gompertz growth trend $y_n$, (11) with errors (see Fig 6):

$$\max_{0 \le n \le 201} |y_n - f(n)| = 525, \quad \text{RMSE} = \sqrt{\frac{1}{202} \sum_{n=0}^{201} (y_n - f(n))^2} = 160, \quad R^2 = 0.999985.$$

## 4 CONCLUSIONS

In the present paper we showed that the Gompertz curve can be approximated, with high accuracy, by a multilogistic function consisting of three logistic functions. We have shown this using an example of one specific time series having the exact Gompertz growth. Obviously, this applies to any Gompertz curve by linearly replacing the variables.

The question arises whether it is possible to interpret somehow the subsequent components of the multilogistic function (12). It seems that the first component in (12) could be interpreted as a diffusion model in a favorable business environment. The second function, with a negative saturation level, could play the role of an inhibitory influence of business competitors, trying to prevent a

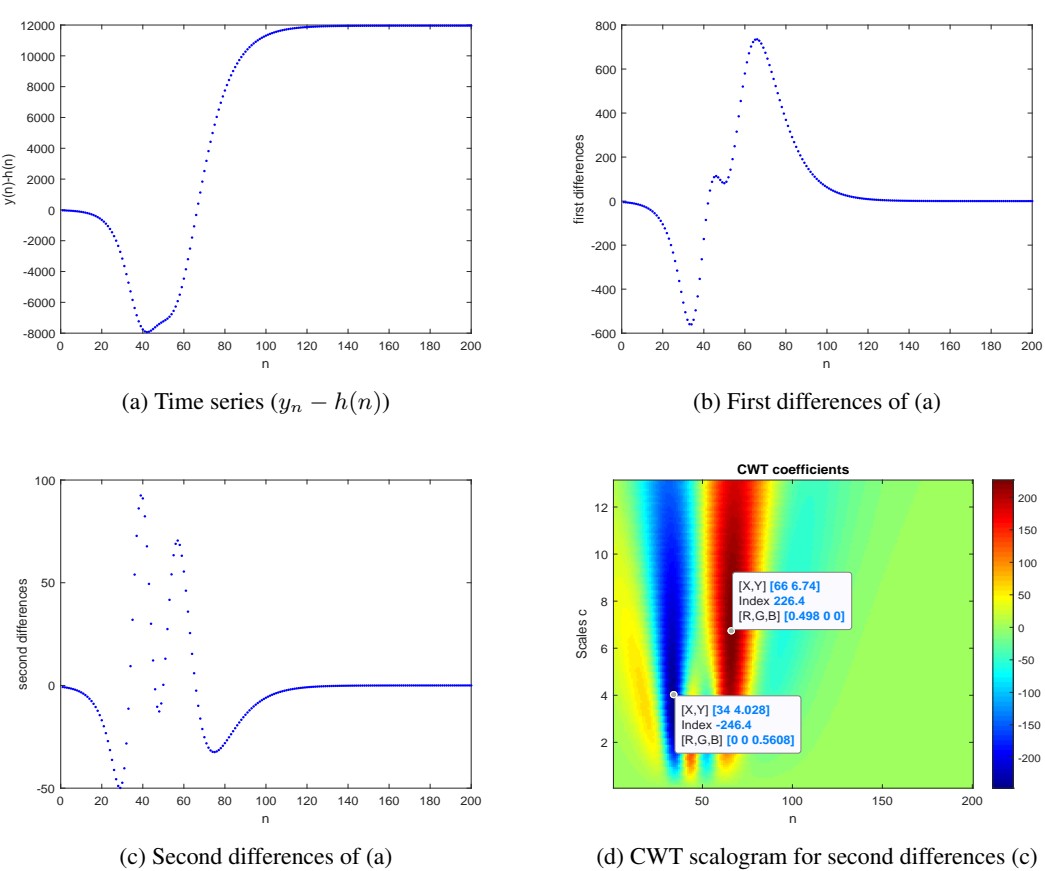

(a) Time series $(y_n - h(n))$

(b) First differences of (a)

(c) Second differences of (a)

(d) CWT scalogram for second differences (c)

Figure 5: Graphs and the CWT scalogram showing the other two logistic waves

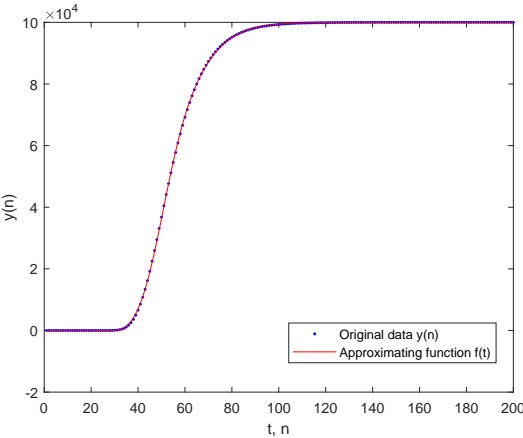

Figure 6: Approximating function $f(t)$, (12)

new, innovative product from entering the market. This last function can be interpreted as a kind of strengthening function (boost function) that appears after the role of the competitors is effectively limited. All these interactions cause the appearance of the Gompertz curve.

## Conflict of Interests

The author declares no conflict of interests related to the submitted manuscript.

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
