# OpenReview forum: "Approximation of the Gompertz trend with a multilogistic function"
_ICLR.cc/2026/Conference — ICLR 2026 Conference Withdrawn Submission_

### Official Review · Reviewer_y4ao · 2025-10-27

**Soundness:** 3
**Presentation:** 3
**Contribution:** 1
**Rating:** 2
**Confidence:** 4

**Summary:**

The paper proposes approximating the Gompertz function by a sum of three logistic (sigmoid) functions. The parameters of these logistic components are identified using the Continuous Wavelet Transform with logistic wavelets.
The approximation achieves a very close numerical approximation (Rsquared approx 0.999985) for a specific Gompertz trend with fixed parameters. A potential interpretation of the three components as representing different growth phases (acceleration, inhibition, and reinforcement) is proposed. No theoretical proofs or general results are provided. The paper demonstrates the idea empirically on one synthetic example.

**Strengths:**

1. Using logistic wavelets to approximate a Gompertz trend is, as dar as I can see, original.
2. The exposition is well structured and easy to follow.
3. I agree that the result that three logistic terms yield such a high-quality approximation could be useful for applied modeling.
4. The approach might extend to other S-shaped growth models or data-driven growth processes. So this example could be the basis for a more comprehensive project.

**Weaknesses:**

1. The scope is limited. Only one specific Gompertz function is tested. The method’s behavior for general parameters or noisy data is not explored. It is vaguely claimed that the approach generalizes, but a formal statement is missing.
2. No mathematical guarantees are given, e.g., no proof that three logistic terms are needed for such an error, or, more interestingly, how the approximation error behaves as more terms are added.
3. It is unclear if the wavelet based approach finds the best three teem approximation.
4. The “interpretation” of the three logistic waves (growth, inhibition, boost) is speculative rather than analytically grounded.
5. No concrete application, where the three term approximation is used to some advantage, is presented.
6. The work seems too narrow and descriptive for a top-tier conference like ICLR.

**Questions:**

1. Can the authors provide theoretical guarantees or error bounds for approximating a Gompertz function with n logistic terms?
2. Is the three-term representation optimal in some sense?
3. How does the approximation quality scale with the number of logistic components? (For more than 3.)
4. Can the method be generalized to Gompertz functions with arbitrary parameters or to noisy empirical data?
5. What would be the advantage of the wavelet-based parameter identification compared to a direct regression fit (e.g., LASSO)?
6. Are there practical applications or case studies where this method provides a measurable advantage?

---

### Official Review · Reviewer_2HTS · 2025-10-29

**Soundness:** 3
**Presentation:** 3
**Contribution:** 1
**Rating:** 0
**Confidence:** 5

**Summary:**

The paper deals with showing that the Gompertz trend can be approximated with high accuracy
by a sum of three logistic functions (multilogistic function), and utilize second-order logistic wavelets to estimate
the parameters of the logistic function.

**Strengths:**

The paper would have had a strength if the contributions were indeed novel, however it has been long known that a Gompertz function is itself a logistic function, so estimating it with a sum of other logistic functions is already well known, routine and doesn't tell us anything new.

I strongly believe the authors of this paper are not human.

**Weaknesses:**

The paper would have had a strength if the contributions were indeed novel, however it has been long known that a Gompertz function is itself a logistic function, so estimating it with a sum of other logistic functions is already well known, routine and doesn't tell us anything new.

I strongly believe the authors of this paper are not human.

**Questions:**

Nil

**Details Of Ethics Concerns:**

I stand to be corrected but strongly feel this paper is LLM generated.

---

> ### Author Response · Authors · 2025-11-12
>
> Dear Reviewer 2HTS,
>
> It's not true that my paper was written by an LLM. I'm sending a link to Arxiv,
> https://arxiv.org/abs/2405.12984
> where you'll find this paper submitted in early 2024. I didn't submit it to any journal or conference until ICLR 2026.
>
> Regards

---

### Official Review · Reviewer_VQpU · 2025-10-30

**Soundness:** 3
**Presentation:** 2
**Contribution:** 2
**Rating:** 2
**Confidence:** 2

**Summary:**

This paper provides a demonstration that a Gompertz curve can be approximated, with high accuracy, by a multilogistic function consisting of three logistic functions.

**Strengths:**

The main strength of the paper is its succinctness and depth of presentation.

**Weaknesses:**

The main weakness of of the appear is its possible lack of relevance to the ICLR community. Although there are uses of Gompertz curves in machine learning the authors make no effort to draw out these connections or illustrate the relevance of their work to a machine learning community.

**Questions:**

What is the relevance of this work to the machine learning community?

---

### Note · Authors · 2025-11-12

**Comment:**

Thank you very much to the Reviewers

**Withdrawal Confirmation:**

I have read and agree with the venue's withdrawal policy on behalf of myself and my co-authors.